# BEYOND INFLUENCE: DECOUPLED REPRESENTATION LEARNING FOR DYNAMIC GRAPH ANOMALY DETECTION

## ABSTRACT

Anomaly detection in dynamic graphs is essential for safeguarding complex systems such as social, financial, and communication networks. A fundamental challenge lies in the entanglement between node influence and anomaly signals. Node influence—measured by metrics such as PageRank—fluctuates naturally over time, yet existing methods often conflate these benign variations with anomalous behaviors, leading to false alarms or missed detections. This paper proposes DIDAN, a framework that distinguishes influence dynamics from true anomalies by separating influence-related and anomaly-related features in dynamic graphs. DIDAN integrates three components: (1) a **Temporal Information Propagator** that learns stable node representations by modeling local and global temporal dependencies; (2) an **Anomaly Feature Synthesizer** that alleviates severe class imbalance by generating diverse synthetic anomalies with a flow-based model; and (3) an **Adversarial Influence-Decoupled Detector** that enforces decoupling through adversarial training. Experiments on multiple real-world dynamic graph benchmarks show that DIDAN consistently outperforms state-of-the-art methods, improving detection accuracy, robustness, and adaptability. Notably, ROC-AUC scores increased by 5.71%, 27.73%, and 1.91% on the Wikipedia, Reddit, and AL-PHA datasets, respectively, highlighting the effectiveness of influence decoupling and anomaly augmentation in dynamic graph anomaly detection.

## 1 INTRODUCTION

Anomaly detection in dynamic graphs is of fundamental importance for ensuring the security, stability, and reliability of complex real-world systems such as financial transaction networks, communication infrastructures, and online social platforms Kim et al. (2024); Li et al. (2023). Detecting abnormal nodes or edges enables applications ranging from fraud prevention to intrusion detection. However, the dynamic and evolving nature of graphs poses significant challenges: nodes naturally vary in their connectivity and activity over time, and naive methods may misinterpret these benign variations as anomalies. As networks grow and adapt, nodes and edges frequently undergo natural fluctuations in connectivity, interaction intensity, and activity patterns. Such variations often arise from role dynamics or structural evolution driven by mechanisms like preferential attachment Qiao et al. (2025). Consequently, naive anomaly detection approaches that rely solely on deviations in node connectivity or embedding trajectories may misinterpret benign variations as anomalies, leading to high false positive rates. At the same time, subtle but genuinely anomalous behaviors may remain hidden within the complex temporal dynamics, resulting in missed detections.

Existing methods for dynamic graph anomaly detection typically rely on learning dynamic node representations and identifying deviations as anomalies. Representative approaches include TGAT (Xu et al., 2020a), which leverages self-attention with time encoding to model temporal dependencies; GDN (Ding et al., 2021), which uses a limited number of labeled anomalies to guide representation learning; SAD (Tian et al., 2023a), integrating a memory bank with pseudo-label contrastive learning to exploit large unlabeled graph streams; TADDY (Liu et al., 2021), which encodes spatial-temporal dependencies via a transformer model; and MAMF (Hong et al., 2025), which employs GAN-based anomaly augmentation combined with meta-learning to handle concept drift. Despite their effectiveness in capturing temporal and structural patterns, these methods share a common lim-

itation: they often misinterpret naturally evolving node influence as anomalies. Nodes with high or rapidly changing influence—quantified, for example, by PageRank (Gleich, 2015)—tend to exhibit larger embedding shifts over time. Since deviation-based detectors use embedding changes as a proxy for abnormality, normal variations in node influence can be mistakenly flagged as anomalies, while truly abnormal nodes with smaller embedding changes may be overlooked. Here, PageRank is used solely to quantify node importance over time and should not be interpreted as influence; the problem arises from treating embedding deviations as anomaly indicators, which conflates normal influence fluctuations with genuine anomalies.

From this observation, we identify two core challenges for robust dynamic graph anomaly detection. **(1) decoupling influence variations from anomalies.** Node influence changes are not anomalies themselves, yet they can dominate node representations and mislead detectors. Separating these effects from true anomaly features is therefore essential. **(2) Data imbalance under dynamic influence.** Anomalies are inherently rare in real-world graphs, and natural variations in node influence can create numerous pseudo-anomalous cases, exacerbating class imbalance and making rare anomalies even harder to detect.

To address these challenges, we propose **DIDAN** (Dynamic Influence-Decoupled Anomaly Network), a novel framework that explicitly separates node influence-related variations from anomaly-related representations. DIDAN integrates three ideas: a temporal propagation mechanism to capture stable and expressive node embeddings, an anomaly synthesizer to generate diverse synthetic anomalies and mitigate imbalance, and an adversarial influence-Decoupled Detector to suppress the interference of dynamic influence variations. Our main contributions are summarized as follows:

- We introduce **DIDAN**, a novel framework for dynamic graph anomaly detection that explicitly decouples node influence variations from anomaly-related features, addressing the conflation of natural influence fluctuations with abnormal behaviors.

- We design a unified framework that integrates a **Temporal Information Propagator (TIP)** to capture local and global temporal dependencies for stable and expressive node embeddings, an **Anomaly Feature Synthesizer (AFS)** to generate realistic synthetic anomalies and mitigate class imbalance, and an **Adversarial Influence-Decoupled Detector (AIDD)** that uses adversarial learning to separate influence-related variations from anomaly signals, reducing false positives and missed detections.

- We conduct extensive experiments on multiple real-world dynamic graph datasets, demonstrating that DIDAN achieves state-of-the-art performance in terms of accuracy, robustness, and adaptability. Notably, the ROC-AUC scores on the Wikipedia, Reddit, and ALPHA datasets improved by 5.71%, 27.73, and 1.91%, respectively.

## 2 RELATED WORK

### 2.1 INFLUENCE PROPAGATION IN DYNAMIC NETWORKS

The study of influence propagation in dynamic networks has garnered significant attention, with various models developed to understand and optimize influence spread over time. Wang et al. (2015) introduced a latent influence propagation model that leverages latent features to simulate influence diffusion on dynamic social networks. Yalavarthi & Khan (2018) proposed a generalized framework for efficient local updates, enabling rapid adjustment of top-k influencers as network structures evolve. Addressing negative influence mitigation, Wu et al. (2019) developed algorithms to minimize dynamic rumor impact by exploiting community structures. Ge et al. (2020) introduced multi-topic influence models (MTL-IC and SPM-EE) that account for complex user interests and temporal topic evolution. Li et al. (2021b) contributed a dynamic influence maximization algorithm based on cohesive entropy (DEIM), emphasizing local aggregation effects and network dynamics. Wang & Zhao (2021) proposed the TPP-DA method for dynamic topic propagation prediction, Li et al. (2021a) examined the reciprocal relationship between influence propagation and network structure using an extended Linear Threshold model. Liu et al. (2022) advanced influence maximization with entropy-based Linear Threshold models, and Jiang et al. (2023) employed deep reinforcement learning for rumor influence minimization. Wu et al. (2023) proposed the Influence SubGraph Propagation (ISGP) method for accurate node influence estimation in dynamic networks.

## 2.2 FEATURE DECOUPLING IN ANOMALY DETECTION

Feature decoupling has emerged as a significant approach in enhancing anomaly detection within dynamic graph environments. Wang et al. (2021) introduced a framework that decouples representation learning from classification using self-supervised learning and Deep Cluster Infomax (DCI) scheme. Zhou et al. (2024) proposed a lightweight model that integrates GNNs with knowledge distillation, extracting structural and traffic features separately using GAT and MLP techniques. Barros et al. (2021) provided a taxonomy of dynamic graph embedding approaches, including matrix factorization, deep learning, and temporal point processes. Cai et al. (2021) developed the StrGNN model that captures structural and temporal features through graph convolution operations. Kim et al. (2025) introduced an innovative approach where timestamps are modeled as distinct nodes, explicitly capturing temporal dependencies.

While these studies have contributed valuable methodologies, existing models still face challenges in handling the complex nature of node interactions, particularly in addressing interference caused by dynamic influence changes. Most approaches focus on separating specific feature types but overlook the dynamic variations of node influence over time, which can severely interfere with anomaly detection performance.

In our work, we propose **DIDAN**, which explicitly addresses the interference of dynamic influence changes through a dual-channel design that decouples influence-related features from anomaly-related features. Our framework consists of three modules: the **Temporal Information Propagator**, **Anomaly Feature Synthesizer**, and **Adversarial Influence-Decoupled Detector**, each designed to tackle specific challenges in dynamic graph anomaly detection.

## 3 METHODOLOGY

In this section, we present the overall architecture of our proposed framework, **DIDAN** (Dynamic Influence-decoupled Anomaly Network), which is specifically designed to address the challenges posed by temporal variations in node influence and class imbalance in dynamic graphs. DIDAN comprises three unified modules, each targeting a distinct challenge identified in the introduction: influence decoupling and anomaly detection under dynamic influence. The overall architecture of DIDAN is illustrated in Figure 1. The following sections provide detailed descriptions of each module, including theoretical foundations and implementation details.

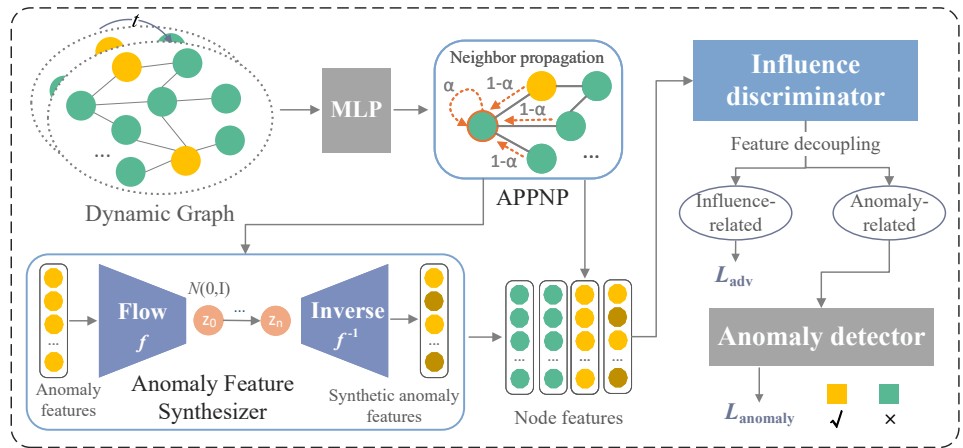

Figure 1: Overall architecture of the DIDANframework.

## 3.1 OVERVIEW OF DIDAN ARCHITECTURE

The DIDAN framework is designed to address two core challenges in dynamic graph anomaly detection: (1) learning high-expressiveness node representations that capture both local and global temporal structure, and (2) decoupling influence-related features from anomaly-related features to

suppress interference from dynamic influence variations. Formally, let $\mathcal{G} = \{G^{(1)}, G^{(2)}, \ldots, G^{(T)}\}$ denote a sequence of dynamic graphs, where $G^{(t)} = (\mathcal{V}^{(t)}, \mathcal{E}^{(t)})$ represents the graph at time $t$, with node set $\mathcal{V}^{(t)}$ and edge set $\mathcal{E}^{(t)}$. Each node $v \in \mathcal{V}^{(t)}$ is associated with a feature vector $\mathbf{x}_v^{(t)} \in \mathbb{R}^d$, and the feature matrix for all nodes at time $t$ is denoted as $\mathbf{X}^{(t)} \in \mathbb{R}^{N \times d}$, where $N = |\mathcal{V}^{(t)}|$ is the number of nodes and $d$ is the feature dimension. The normalized adjacency matrix with self-loops is denoted as $\hat{\mathbf{A}} \in \mathbb{R}^{N \times N}$.

The three core components of DIDAN are as follows:

- **Temporal Information Propagator (TIP)**: This module captures both local and global temporal dependencies through multi-step neighbor propagation. By providing stable and expressive node representations that are less biased by transient influence fluctuations, TIP forms a reliable foundation for subsequent influence decoupling.

- **Anomaly Feature Synthesizer (AFS)**: To mitigate the severe class imbalance inherent in real-world dynamic graphs, AFS employs a flow-based generative model to produce high-quality synthetic anomalies, ensuring sufficient anomaly samples for effective training.

- **Adversarial Influence-Decoupled Detector (AIDD)**: This module explicitly separates influence-related features from anomaly-related features using a projection–back-projection architecture with adversarial training. By decoupling the influence signals captured by node metrics such as degree, betweenness, and PageRank, AIDD enables accurate detection of true anomalies under fluctuating node influence.

## 3.2 Temporal Information Propagator (TIP)

The TIP module generates stable and expressive node representations by propagating initial features through the graph using APPNP (Gasteiger et al., 2018). Let $\mathbf{X}^{(t)} \in \mathbb{R}^{N \times d}$ denote the input node features at time step $t$. The initial embeddings are obtained via a multi-layer perceptron (MLP):

$$\mathbf{H}_0^{(t)} = \text{MLP}(\mathbf{X}^{(t)}), \tag{1}$$

and updated recursively as

$$\mathbf{H}_k^{(t)} = (1 - \alpha)\hat{\mathbf{A}}\mathbf{H}_{k-1}^{(t)} + \alpha\mathbf{H}_0^{(t)}, \quad k = 1, 2, \ldots, K, \tag{2}$$

where $\alpha \in (0, 1)$ is the teleport probability.

By iteratively propagating features while retaining a fraction of the original node information, TIP produces embeddings that are smooth across the graph yet discriminative for individual nodes. We formally prove in Appendix C.1 that this iterative propagation converges to a unique fixed point (Theorem 1) and that the final embeddings preserve contributions from the original features, ensuring node individuality (Corollary 2).

## 3.3 Anomaly Feature Synthesizer (AFS)

The AFS module alleviates the class imbalance problem by generating high-quality synthetic anomaly features, enabling the detector to learn robust decision boundaries even with scarce anomaly samples.

Let $\mathcal{D}_a = \{\mathbf{x}_i\}_{i=1}^{N_a}$ denote the set of observed anomaly node features. We employ a flow-based generative model $f_\theta : \mathbb{R}^d \to \mathbb{R}^d$ that is invertible and differentiable, mapping latent variables $\mathbf{z} \sim \mathcal{N}(\mathbf{0}, \mathbf{I})$ to the feature space:

$$\mathbf{x} = f_\theta^{-1}(\mathbf{z}), \quad \mathbf{z} \sim \mathcal{N}(\mathbf{0}, \mathbf{I}), \tag{3}$$

with likelihood

$$p_\theta(\mathbf{x}) = p_{\mathbf{z}}(f_\theta(\mathbf{x})) \left| \det \frac{\partial f_\theta}{\partial \mathbf{x}} \right|, \tag{4}$$

and training objective

$$
\begin{aligned}
\mathcal{L}_{\text{flow}}(\theta) &= -\frac{1}{N_a} \sum_{i=1}^{N_a} \log p_\theta(\mathbf{x}_i) \\
&= -\frac{1}{N_a} \sum_{i=1}^{N_a} \left[ \log p_{\mathbf{z}}(f_\theta(\mathbf{x}_i)) + \log \left| \det \frac{\partial f_\theta}{\partial \mathbf{x}_i} \right| \right].
\end{aligned}
\tag{5}
$$

Synthetic anomalies $\mathbf{x}_{\text{gen}}$ are generated by sampling $\mathbf{z} \sim \mathcal{N}(\mathbf{0}, \mathbf{I})$ and applying $\mathbf{x}_{\text{gen}} = f_\theta^{-1}(\mathbf{z})$. We formally prove in Appendix C.1 (Theorem 3) that the likelihood is exact and the generation process is bijective. As a consequence, any generated sample $\mathbf{x}_{\text{gen}}$ is guaranteed to follow the learned anomaly distribution, ensuring statistical consistency and well-defined features for training.

By augmenting the training set with $\{\mathbf{x}_{\text{gen}}\}$, the AFS module effectively mitigates class imbalance, allowing the anomaly detector to learn accurate boundaries between normal and abnormal nodes. These theoretical guarantees (Appendix C.1, Theorem 3) justify the use of flow-based generation and support the robustness of DIDAN in dynamic graphs with scarce anomalies.

## 3.4 ADVERSARIAL INFLUENCE-DECOUPLED DETECTOR (AIDD)

The AIDD module aims to decouple node embeddings into components correlated with normal influence variations and components indicative of genuine anomalies. This ensures that the anomaly detector focuses on abnormal behaviors rather than benign fluctuations in node influence.

**Design Assumption:** We assume that temporal variations in a node's PageRank primarily reflect normal influence changes. Under this assumption, we aim to extract embedding components correlated with PageRank and remove them from the anomaly detection features.

**Projection–Back-Projection:** Let $\mathbf{h} \in \mathbb{R}^d$ be the embedding obtained from TIP. We introduce a linear projection $\mathbf{W}_p$ and a back-projection $\mathbf{W}_b$ to isolate PageRank-correlated components:

$$
\mathbf{h}_{\text{pr}} = \mathbf{W}_p \mathbf{h}, \quad \mathbf{h}_{\text{main}} = \mathbf{h} - \mathbf{W}_b \mathbf{h}_{\text{pr}}
\tag{6}
$$

**Rationale:** The projection $\mathbf{W}_p$ maps the embedding into a "PageRank information subspace," capturing components most correlated with normal influence changes. The back-projection $\mathbf{W}_b$ removes these components from $\mathbf{h}_{\text{main}}$, so that the features used for anomaly detection are less sensitive to normal PageRank fluctuations. This design separates the embedding space into influence-correlated and anomaly-related subspaces.

**Adversarial Mechanism for decoupling:** While projection/back-projection isolates the PageRank-correlated component, it does not guarantee that $\mathbf{h}_{\text{main}}$ is fully decoupled. We employ an adversarial training strategy using a PageRank predictor $D_{\text{PR}}$ and a gradient reversal layer (GRL). The adversarial loss is:

$$
\mathcal{L}_{\text{adv}} = \text{MSE}\Big( D_{\text{PR}}(\text{GRL}(\mathbf{h}_{\text{pr}})), y^{\text{PR}} \Big)
\tag{7}
$$

The total loss for training AIDD is:

$$
\mathcal{L} = \mathcal{L}_{\text{anomaly}} + \lambda_{\text{adv}} \mathcal{L}_{\text{adv}}
\tag{8}
$$

**Mechanism:** The GRL ensures that $\mathbf{h}_{\text{main}}$ receives gradients that penalize information predictive of PageRank, while $\mathbf{h}_{\text{pr}}$ is optimized to retain PageRank information. At equilibrium, $\mathbf{h}_{\text{main}}$ contains minimal PageRank-related information, achieving effective decoupling. Here, $\lambda_{\text{adv}}$ is a trade-off hyperparameter that balances anomaly detection loss and adversarial loss.

**Limitations:** This decoupling is specific to the PageRank metric. Other forms of influence or node metrics may still leak into $\mathbf{h}_{\text{main}}$. Extending this mechanism to multiple influence measures requires additional projections and adversarial discriminators.

**Integration with DIDAN:** Finally, TIP, AFS, and AIDD are trained jointly. TIP provides expressive embeddings, AFS augments the training set with synthetic anomalies, and AIDD decouples embeddings to produce robust anomaly predictions. Appendix C.1 provides a formal proof (Theorem 4) that the adversarial mechanism ensures decoupling under ideal conditions.

## 3.5 ALGORITHMIC PROCEDURE

The DIDAN framework performs dynamic graph anomaly detection through a three-stage workflow, integrating temporal feature propagation, anomaly feature augmentation, and influence-aware decoupling. Specifically:

1. **Temporal Embedding Extraction (TIP):** For each time step, node features are propagated across the graph using a personalized propagation scheme, capturing temporal dependencies and evolving structural patterns. The resulting embeddings encode both the historical and local neighborhood information of each node.

2. **Anomaly Feature Synthesis (AFS):** To address the scarcity of anomalous nodes, known anomaly embeddings are used to train a flow-based generator. The generator produces synthetic anomaly features that augment the training data, improving the model's exposure to diverse anomalous patterns and mitigating class imbalance.

3. **Influence decoupling (AIDD):** Node embeddings are projected onto an influence subspace to separate benign structural influences (e.g., PageRank-like effects) from genuine anomaly signals. Adversarial training ensures that the decoupled representations focus on anomaly-relevant information, reducing false positives and enhancing detection robustness.

This modular design allows DIDAN to jointly capture temporal dynamics, enrich anomaly representations, and suppress confounding influences. For full implementation details and step-by-step training pseudocode, see Appendix D.

## 3.6 TIME COMPLEXITY ANALYSIS

Here, $N$ is the number of nodes, $E$ is the number of edges, $d$ is the feature dimension, $K$ is the number of propagation steps, $T$ is the number of time steps, $L$ is the number of layers in the flow model, $B$ is the batch size, $S$ is the number of synthetic anomaly samples, $M$ is the number of influence metrics/discriminators, and $E_f$ is the number of epochs for flow model training.

We analyze the time complexity of each module in the DIDAN framework as follows:

**1. Temporal Information Propagator (TIP):**

For each time step $t$, the main computational cost comes from the APPNP propagation. Let $N$ be the number of nodes, $E$ the number of edges, $d$ the feature dimension, and $K$ the number of propagation steps. The complexity per step is $O(Ed)$ for sparse matrix multiplication, and the total for $K$ steps is $O(KEd)$. Including the initial MLP, the overall complexity per time step is $O(Nd^2 + KEd)$. For $T$ time steps, the total is $O(T(Nd^2 + KEd))$.

**2. Anomaly Feature Synthesizer (AFS):**

Assuming the flow-based model (e.g., RealNVP) has $L$ layers and batch size $B$, the forward and inverse pass per sample is $O(Ld^2)$. For $S$ synthetic samples generated, the total complexity is $O(SLd^2)$. Training the flow model on $M$ anomaly samples for $E_f$ epochs is $O(E_f MLd^2)$.

**3. Adversarial Influence-Decoupled Detector (AIDD):**

For each batch, the projection and back-projection are $O(d^2)$ per node. The adversarial discriminators (assuming $M$ influence metrics) add $O(Md^2)$ per node. For $B$ nodes per batch, the total is $O(BMd^2)$. The anomaly classifier is also $O(Bd^2)$ per batch.

**Overall Complexity:**

The overall time complexity per epoch is dominated by the TIP module (graph propagation) and the AIDD module (adversarial training). For a dynamic graph with $T$ time steps, $N$ nodes, $E$ edges, feature dimension $d$, $K$ propagation steps, and $M$ influence metrics, the total complexity per epoch

Table 1: Performance comparison on Wikipedia, Reddit, and UCI datasets.

| Method | Wikipedia | | | Reddit | | | UCI | | |
|---|---|---|---|---|---|---|---|---|---|
| | ROC-AUC | F1-score | AUPR | ROC-AUC | F1-score | AUPR | ROC-AUC | F1-score | AUPR |
| TGAT | 0.7576 | 0.4995 | 0.0251 | 0.6222 | 0.4998 | 0.0020 | 0.9491 | 0.4985 | 0.1258 |
| DOMINANT | 0.6707 | 0.1022 | 0.0845 | 0.6101 | 0.2281 | 0.1602 | 0.5233 | 0.0805 | 0.1912 |
| DONE | 0.6486 | 0.0839 | 0.0775 | 0.5690 | 0.2144 | 0.1353 | 0.4993 | 0.0850 | 0.1848 |
| CONAD | 0.6698 | 0.1022 | 0.0845 | 0.6119 | 0.2300 | 0.1602 | 0.5225 | 0.0805 | 0.1913 |
| AnomalyDAE | 0.6706 | 0.0996 | 0.0831 | 0.5666 | 0.2222 | 0.1444 | 0.4912 | 0.0805 | 0.1835 |
| SAD | 0.8641 | 0.4995 | 0.0181 | 0.6880 | 0.4998 | 0.0027 | 0.9223 | 0.4985 | 0.1746 |
| MAMF | 0.9355 | 0.8307 | 0.7507 | 0.7221 | 0.6425 | 0.6053 | 0.9735 | 0.9741 | 0.9725 |
| DIDAN(Ours) | **0.9926** | **0.9873** | **0.9953** | **0.9994** | **0.9963** | **0.9996** | **0.9910** | **0.9828** | **0.9928** |

Table 2: Performance comparison on EU-Core1/3, OTC, and ALPHA datasets.

| Method | EU-Core1 | | | EU-Core3 | | | OTC | | | ALPHA | | |
|---|---|---|---|---|---|---|---|---|---|---|---|---|
| | ROC-AUC | F1-score | AUPR | ROC-AUC | F1-score | AUPR | ROC-AUC | F1-score | AUPR | ROC-AUC | F1-score | AUPR |
| TGAT | 0.4475 | 0.4986 | 0.0057 | 0.5558 | 0.4972 | 0.0980 | 0.6730 | 0.4770 | 0.1845 | 0.7542 | 0.4836 | 0.1427 |
| DOMINANT | 0.5282 | 0.1162 | 0.2214 | 0.5856 | 0.0995 | 0.2123 | 0.6227 | 0.2518 | 0.4129 | 0.6823 | 0.3289 | 0.4152 |
| DONE | 0.5138 | 0.1328 | 0.2247 | 0.5535 | 0.0724 | 0.2019 | 0.6347 | 0.2710 | 0.4350 | 0.6843 | 0.3237 | 0.4006 |
| CONAD | 0.5255 | 0.1162 | 0.2199 | 0.5868 | 0.0995 | 0.2128 | 0.6250 | 0.2556 | 0.4137 | 0.6816 | 0.3368 | 0.4107 |
| AnomalyDAE | 0.4993 | 0.1162 | 0.2170 | 0.5835 | 0.0905 | 0.2063 | 0.3941 | 0.0347 | 0.1694 | 0.4962 | 0.0474 | 0.1436 |
| SAD | 0.5361 | 0.4972 | 0.0170 | 0.9080 | 0.4972 | 0.1555 | 0.7173 | 0.4770 | 0.1997 | 0.7574 | 0.4836 | 0.1414 |
| MAMF | 0.9573 | 0.9455 | 0.9258 | 0.9403 | 0.8748 | 0.8381 | 0.9415 | 0.9284 | 0.9074 | 0.9355 | 0.9293 | 0.9320 |
| DIDAN(Ours) | **0.9604** | **0.9457** | **0.9601** | **0.9558** | **0.9407** | **0.9644** | **0.9516** | **0.9362** | **0.9668** | **0.9546** | **0.9317** | **0.9679** |

is:

$$O\left(T(Nd^2 + KEd) + BMd^2 + SLd^2\right) \tag{9}$$

where $B$ is the batch size and $S$ is the number of synthetic samples generated by AFS.

In practice, the method is efficient for sparse graphs ($E \ll N^2$) and moderate feature dimensions, and the modules can be parallelized to further accelerate training.

## 4 EXPERIMENTS

### 4.1 EXPERIMENTAL SETTINGS

*(a) Datasets.* We assess DIDAN on diverse real-world dynamic graph datasets: **Wikipedia** (hyperlink evolution) (Kumar et al., 2019), **Reddit** (user interactions) (Kumar et al., 2019), **EU-Core1/3** (collaboration networks) (Guo et al., 2022), **OTC/ALPHA** (Bitcoin transactions) (Kumar et al., 2016), and **UCI** (Message Network) (Zheng et al., 2019). All datasets exhibit severe class imbalance with anomalous nodes constituting less than 5% of the total.

*(b) Baselines.* We compare against seven representative methods: **TGAT** (Xu et al., 2020b) (temporal attention networks), **DOMINANT** (Ding et al., 2019) (structure/attribute reconstruction), **DONE** (Bandyopadhyay et al., 2020) (GCN with temporal autoencoders), **CONAD** (Xu et al., 2022) (contrastive learning with temporal augmentation), **AnomalyDAE** (Fan et al., 2020) (deep autoencoder), **SAD** (Tian et al., 2023b) (semi-supervised detection), and **MAMF** (Hong et al., 2025) (multitask meta-learning). These baselines cover a wide range of paradigms, including temporal modeling, reconstruction-based, contrastive, semi-supervised, and meta-learning approaches.

*(c) Evaluation Metrics.* We adopt four metrics: **ROC-AUC** (ranking ability) (Huang & Ling, 2005), **F1-score** (precision-recall balance) (Huang et al., 2015), and **AUPR** (imbalanced data performance) (Zhou, 2023).

*(d) Implementation Details.* Each dataset is split into five temporal segments (last for testing, first four for training/validation). Node embeddings dimension is 128, batch size is 100, $\lambda_{\text{adv}}$ is set to 1. We train the model using mini-batches of 100 nodes, with the Adam optimizer (Bock & Weiß, 2019) and an initial learning rate of 0.0001. Experiments run on NVIDIA GeForce RTX 4060 GPU with 20 runs for statistical reliability.

### 4.2 PERFORMANCE COMPARISON

We benchmark DIDAN against seven representative baselines: TGAT Xu et al. (2020b), DOMINANT Ding et al. (2019), DONE Bandyopadhyay et al. (2020), CONAD Xu et al. (2022), Anoma-

lyDAE Fan et al. (2020), SAD Tian et al. (2023b), and MAMF Hong et al. (2025), across seven benchmark datasets (Wikipedia, Reddit, UCI, EU-Core1/3, OTC, ALPHA). As shown in Tables 1 and 2, DIDAN achieves the best or near-best performance on all datasets and metrics. Notably, it significantly outperforms strong baselines such as MAMF and SAD, especially on large-scale graphs with complex dynamics. This demonstrates the scalability and robustness of our approach. Overall, the consistent gains highlight the effectiveness of our three key components—TIP, AFS, and AIDD—in learning discriminative and generalizable anomaly representations across diverse dynamic graph scenarios.

### 4.3 ABLATION STUDY

We study the impact of TIP propagation and influence decoupling in DIDAN via two variants: (i) **w/o APPNP**, replacing APPNP with GCN; (ii) **w/o ID**, disabling AIDD. Experiments use 128-d embeddings, batch size 100, 20 runs.

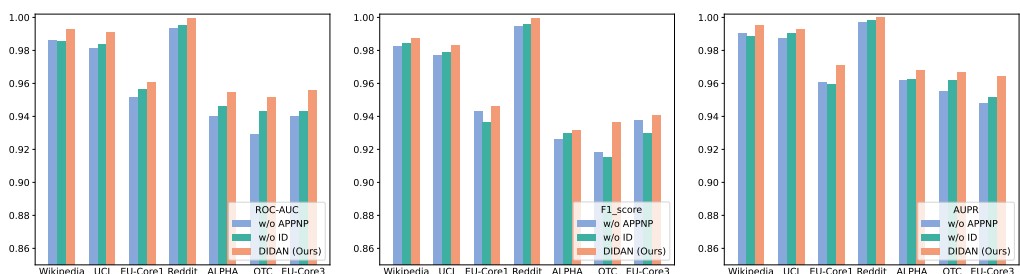

Figure 2: Ablation study: removing key components degrades performance, highlighting their importance. Left: ROC-AUC; Middle: F1-score; Right: AUPR.

Results show that removing APPNP or disabling AIDD clearly reduces performance, confirming that both TIP propagation and influence decoupling are critical for robust anomaly detection.

### 4.4 PARAMETER SENSITIVITY ANALYSIS

We investigate the sensitivity of DIDAN to the learning rate, varying it over the range $\{1e{-}6, 5e{-}6, 1e{-}5, 5e{-}5, 1e{-}4, 5e{-}3\}$. All other hyperparameters are kept fixed.

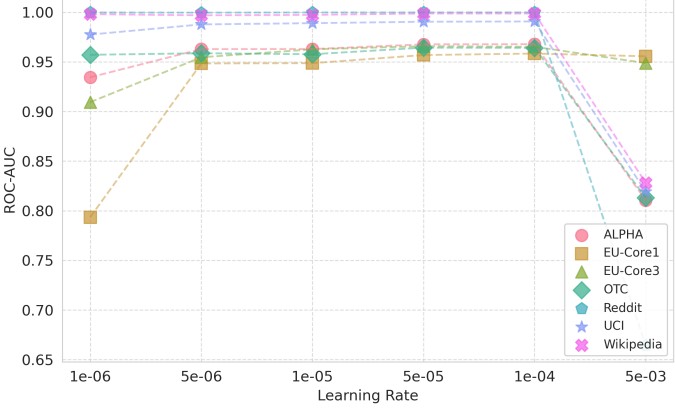

Figure 3: Effect of learning rate on model performance. Performance increases from $1e{-}6$ to $5e{-}6$, then plateaus, and declines for $1e{-}4$ to $5e{-}3$.

As shown in Figure 3, performance first improves from $1 \times 10^{-6}$ to $5 \times 10^{-6}$, then stays almost constant between $5 \times 10^{-6}$ and $1 \times 10^{-4}$, and finally declines for learning rates larger than $1 \times 10^{-4}$. Following this trend, we select $1 \times 10^{-4}$ as the default learning rate. This analysis confirms that

DIDAN is relatively stable over a small range of learning rates, but excessively large values can harm training.

## 4.5 EMBEDDING VISUALIZATION

To qualitatively illustrate the effect of influence decoupling (ID) on node representations, we project learned embeddings to three-dimensional space using PCA. Figure 4 shows the UCI dataset as an example, where purple nodes are anomalous and yellow nodes are normal.

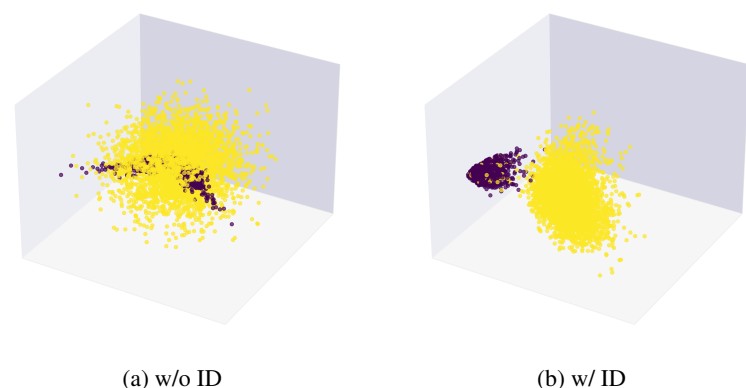

(a) w/o ID          (b) w/ ID

Figure 4: PCA-based 3D visualization of node embeddings on the UCI dataset. Influence decoupling (ID) improves separation between anomalous and normal nodes.

As seen in Figure 4, applying ID clearly increases the separation between anomalous and normal nodes. This demonstrates that AIDD effectively removes influence-related variations, allowing embeddings to focus on genuine anomalies. For completeness, visualizations of embeddings on the other six datasets are provided in Appendix E.

## 5 CONCLUSION

In this paper, we proposed DIDAN, a Dynamic Influence-decoupled Generative Anomaly detector for dynamic graphs. DIDAN effectively addresses two key challenges: extracting robust temporal node representations through the Temporal Information Propagator (TIP), and decoupling influence-related features from anomaly-related features via the Adversarial Influence-Decoupled Detector (AIDD). Additionally, the Anomaly Feature Synthesizer (AFS) alleviates class imbalance by generating high-quality synthetic anomalies. Extensive experiments on real-world dynamic graph datasets demonstrate that DIDAN outperforms state-of-the-art baselines in anomaly detection accuracy and robustness, particularly under severe class imbalance and dynamic influence variations. These results validate the effectiveness of our dual-channel design and the importance of influence decoupling in dynamic graph anomaly detection. In summary, DIDAN provides a practical and effective framework for detecting node-level anomalies in evolving networks, combining robust temporal feature extraction, generative augmentation, and adversarial feature decoupling.

## ETHICS STATEMENT

Our research focuses on anomaly detection in dynamic graphs, with applications in fraud detection, intrusion detection, and monitoring of online platforms. The methods proposed are intended for legitimate and responsible uses, aiming to enhance the security, stability, and reliability of complex systems. We acknowledge that misuse of anomaly detection technologies could lead to privacy concerns or unfair treatment of individuals, and we emphasize that all deployments should comply with relevant laws and ethical guidelines. Our experiments use publicly available datasets, ensuring no private or sensitive user data is exposed.

## REPRODUCIBILITY STATEMENT

We have taken steps to ensure the reproducibility of our results. All datasets used in our experiments are publicly available, and the main text provides detailed descriptions of data preprocessing, model architectures, hyperparameters, and training procedures. The code for our experiments is publicly released and available at https://anonymous.4open.science/r/DIDAN-6D6F, allowing other researchers to directly reproduce our results on the same datasets.

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

## A    USE OF LARGE LANGUAGE MODELS (LLMS)

We made limited use of ChatGPT (GPT-4) in this work. Specifically, it was employed for language polishing (grammar checks and writing style refinement) and providing small suggestions for code debugging. The LLM did not contribute to research conception, methodology, or analysis; all substantive contributions were made by the authors.

## B    NOTATION

Table 3 summarizes the main mathematical symbols used throughout the description of the DIDAN framework. Each symbol is listed with its meaning and, where appropriate, its type or shape. This table serves as a reference for understanding the notations in the methodology section.

Table 3: Symbol Table for DIDANFramework

| Symbol | Description |
|---|---|
| $\mathcal{G}$ | Sequence of dynamic graphs |
| $G^{(t)}$ | Graph at time $t$ |
| $\mathcal{V}^{(t)}$ | Node set at time $t$ |
| $\mathcal{E}^{(t)}$ | Edge set at time $t$ |
| $\mathbf{x}_v^{(t)}$ | Feature vector of node $v$ at time $t$ |
| $\mathbf{X}^{(t)}$ | Feature matrix at time $t$ |
| $K$ | Number of propagation steps (APPNP) |
| $\alpha$ | Teleport (restart) probability (APPNP) |
| $\hat{\mathbf{A}}$ | Normalized adjacency matrix |
| $\mathbf{H}_k^{(t)}$ | Node representations at step $k$ |
| $L$ | Number of layers in flow model |
| $B$ | Batch size |
| $S$ | Number of synthetic anomaly samples |
| $M$ | Number of influence metrics/discriminators |
| $\mathbf{h}$ | Node representation |
| $\mathbf{h}_{\mathrm{inf}}$ | Influence-related feature |
| $\mathbf{h}_{\mathrm{main}}$ | Anomaly-related feature |
| $\mathbf{W}_p, \mathbf{W}_b$ | Projection/back-projection matrices |
| $D_i$ | $i$-th influence discriminator |
| $y_i^{\mathrm{inf}}$ | $i$-th influence label |
| $\mathcal{L}_{\mathrm{anomaly}}$ | Anomaly detection loss |
| $\mathcal{L}_{\mathrm{adv}}$ | Adversarial loss |
| $\lambda_{\mathrm{adv}}$ | Adversarial loss weight |

## C    APPENDIX: THEORETICAL PROOFS FOR DIDAN MODULES

### C.1    PROOF OF TIP CONVERGENCE AND FEATURE PRESERVATION

The Temporal Information Propagator (TIP) iteratively propagates node features across the graph while retaining a fraction of the original features. Although the recursive formulation in Eq. equation 2 is intuitive, it is important to formally establish that this iterative process converges to a unique fixed point and that the final embeddings preserve contributions from the initial features. Without these guarantees, repeated propagation could lead to unstable representations or loss the individuality of nodes, which would undermine the reliability of downstream anomaly detection.

Formally, TIP updates are given by

$$\mathbf{H}_0^{(t)} = \mathrm{MLP}(\mathbf{X}^{(t)}), \tag{10}$$

$$\mathbf{H}_k^{(t)} = (1 - \alpha)\hat{\mathbf{A}}\mathbf{H}_{k-1}^{(t)} + \alpha\mathbf{H}_0^{(t)}, \quad k = 1, 2, \ldots, K, \tag{11}$$

where $\hat{\mathbf{A}}$ is the normalized adjacency matrix with self-loops, and $\alpha \in (0, 1)$.

**Theorem 1** (Convergence of TIP embeddings). *The sequence $\{\mathbf{H}_k^{(t)}\}_{k=0}^{\infty}$ defined in Eq. 11 converges to a unique fixed point:*

$$\mathbf{H}_{\infty}^{(t)} = \alpha(\mathbf{I} - (1 - \alpha)\hat{\mathbf{A}})^{-1}\mathbf{H}_0^{(t)}. \tag{12}$$

*Proof.* Define the error at step $k$ as

$$\mathbf{E}_k = \mathbf{H}_k^{(t)} - \mathbf{H}_{\infty}^{(t)}.$$

From Eq. 11 and the fixed point condition, we have

$$\mathbf{H}_{\infty}^{(t)} = (1 - \alpha)\hat{\mathbf{A}}\mathbf{H}_{\infty}^{(t)} + \alpha\mathbf{H}_0^{(t)} \quad \implies \quad \mathbf{E}_k = (1 - \alpha)\hat{\mathbf{A}}\mathbf{E}_{k-1}.$$

Applying this relation recursively yields

$$\mathbf{E}_k = ((1 - \alpha)\hat{\mathbf{A}})^k\mathbf{E}_0.$$

Since $\hat{\mathbf{A}}$ is row-stochastic, its spectral radius $\rho(\hat{\mathbf{A}}) = 1$, and thus

$$\rho((1 - \alpha)\hat{\mathbf{A}}) = 1 - \alpha < 1.$$

By standard linear algebra results, $\lim_{k \to \infty}((1 - \alpha)\hat{\mathbf{A}})^k = 0$, implying

$$\lim_{k \to \infty} \mathbf{E}_k = 0 \quad \implies \quad \lim_{k \to \infty} \mathbf{H}_k^{(t)} = \mathbf{H}_{\infty}^{(t)}.$$

This completes the proof of convergence. □

**Corollary 2** (Preservation of original features). *The fixed-point embedding $\mathbf{H}_{\infty}^{(t)}$ preserves contributions from the initial node features:*

$$\mathbf{H}_{\infty}^{(t)} = \alpha \sum_{i=0}^{\infty} ((1 - \alpha)\hat{\mathbf{A}})^i\mathbf{H}_0^{(t)}. \tag{13}$$

*Proof.* Unrolling the recursion in Eq. 11 gives

$$\mathbf{H}_k^{(t)} = \alpha \sum_{i=0}^{k-1} ((1 - \alpha)\hat{\mathbf{A}})^i\mathbf{H}_0^{(t)} + ((1 - \alpha)\hat{\mathbf{A}})^k\mathbf{H}_0^{(t)}.$$

Taking the limit $k \to \infty$, the second term vanishes because $\rho((1 - \alpha)\hat{\mathbf{A}}) < 1$, leaving

$$\mathbf{H}_{\infty}^{(t)} = \alpha \sum_{i=0}^{\infty} ((1 - \alpha)\hat{\mathbf{A}})^i\mathbf{H}_0^{(t)},$$

which proves that the final embedding retains contributions from the initial features. □

THEORETICAL JUSTIFICATION FOR THE ANOMALY FEATURE SYNTHESIZER

In this appendix, we provide a formal justification for the use of flow-based generative models in the AFS module. The AFS module generates synthetic anomaly features to mitigate class imbalance in dynamic graph anomaly detection. Since these synthetic features are used for training the anomaly detector, it is crucial to guarantee that they are *well-defined* and *statistically consistent* with the observed anomalies. Without such guarantees, unreliable synthetic samples could bias the learned decision boundaries.

We now formally prove that the likelihood of any anomaly feature is exact and that the generation process is bijective.

**Theorem 3** (Exact likelihood and bijectivity). *Let $f_\theta : \mathbb{R}^d \to \mathbb{R}^d$ be an invertible and differentiable mapping. Then for any $\mathbf{x} \in \mathbb{R}^d$, there exists a unique $\mathbf{z} \in \mathbb{R}^d$ such that $\mathbf{x} = f_\theta^{-1}(\mathbf{z})$, and the probability density*

$$p_\theta(\mathbf{x}) = p_{\mathbf{z}}(f_\theta(\mathbf{x})) \left| \det \frac{\partial f_\theta}{\partial \mathbf{x}} \right| \tag{14}$$

*is exact. Consequently, sampling $\mathbf{z} \sim \mathcal{N}(\mathbf{0}, \mathbf{I})$ and computing $\mathbf{x}_{gen} = f_\theta^{-1}(\mathbf{z})$ produces synthetic anomalies that follow the learned anomaly distribution exactly.*

*Proof.* Since $f_\theta$ is invertible by assumption, for any $\mathbf{x} \in \mathbb{R}^d$, there exists a unique $\mathbf{z} = f_\theta(\mathbf{x})$ such that $\mathbf{x} = f_\theta^{-1}(\mathbf{z})$.

By the multivariate change-of-variables formula for probability densities, if $\mathbf{z}$ has density $p_{\mathbf{z}}(\mathbf{z})$, then the density of $\mathbf{x}$ induced by $f_\theta^{-1}$ is

$$p_\theta(\mathbf{x}) = p_{\mathbf{z}}(f_\theta(\mathbf{x})) \left| \det \frac{\partial f_\theta(\mathbf{x})}{\partial \mathbf{x}} \right|. \tag{15}$$

This formula is exact because:

- $f_\theta$ is differentiable, so the Jacobian $\frac{\partial f_\theta}{\partial \mathbf{x}}$ exists everywhere;

- $f_\theta$ is invertible, so the mapping between $\mathbf{x}$ and $\mathbf{z}$ is one-to-one, ensuring no ambiguity in density transformation.

Furthermore, for any $\mathbf{z} \sim \mathcal{N}(\mathbf{0}, \mathbf{I})$, let $\mathbf{x}_{\text{gen}} = f_\theta^{-1}(\mathbf{z})$. Then by construction, $\mathbf{x}_{\text{gen}}$ is uniquely determined by $\mathbf{z}$, and the distribution of $\mathbf{x}_{\text{gen}}$ is exactly $p_\theta(\mathbf{x})$, due to the change-of-variables formula.

Hence, the generated samples are well-defined, bijective with respect to the latent space, and follow the learned anomaly distribution exactly. $\qquad\square$

**Remarks.** This theorem guarantees that:

1. The log-likelihood of any anomaly feature can be computed exactly, providing statistical consistency during training.

2. The generation process is bijective, ensuring synthetic anomalies faithfully follow the observed anomaly distribution.

These properties justify the theoretical validity of using flow-based models in the AFS module for alleviating class imbalance.

THEORETICAL JUSTIFICATION FOR ADVERSARIAL INFLUENCE-DECOUPLED DETECTOR (AIDD)

In this appendix, we provide a formal discussion and theoretical guarantee for the feature disentanglement in AIDD. The goal is to show that, under ideal optimization, the anomaly-related embedding $\mathbf{h}_{\text{main}}$ becomes invariant to PageRank-correlated variations captured by $\mathbf{h}_{\text{pr}}$.

**Preliminaries** Let $\mathbf{h} \in \mathbb{R}^d$ be the embedding from TIP. The AIDD module decomposes it as

$$\mathbf{h}_{\text{pr}} = \mathbf{W}_p \mathbf{h}, \quad \mathbf{h}_{\text{main}} = \mathbf{h} - \mathbf{W}_b \mathbf{h}_{\text{pr}}, \tag{16}$$

where $\mathbf{W}_p \in \mathbb{R}^{d' \times d}$ and $\mathbf{W}_b \in \mathbb{R}^{d \times d'}$ are learnable matrices. The adversarial objective is to predict the PageRank value $y^{\text{PR}}$ from $\mathbf{h}_{\text{pr}}$ using a discriminator $D_{\text{PR}}$ and a GRL:

$$\mathcal{L}_{\text{adv}} = \text{MSE}(D_{\text{PR}}(\text{GRL}(\mathbf{h}_{\text{pr}})), y^{\text{PR}}). \tag{17}$$

**Feature Decoupling Guarantee** This appendix provides formal guarantees for the feature decoupling mechanism in AIDD. The goal is to show that under ideal conditions, the adversarial training ensures that the anomaly-related embedding $\mathbf{h}_{\text{main}}$ contains minimal information predictive of PageRank, achieving disentanglement.

**Theorem 4** (Feature Decoupling via Adversarial Training). *Let $\mathbf{h} \in \mathbb{R}^d$ be the node embedding obtained from TIP. Define the projection $\mathbf{h}_{pr} = \mathbf{W}_p \mathbf{h}$ and back-projection $\mathbf{h}_{main} = \mathbf{h} - \mathbf{W}_b \mathbf{h}_{pr}$ as in Eq. 6. Let $D_{PR}$ be the PageRank predictor with adversarial loss $\mathcal{L}_{adv}$ defined in Eq. 7.*

*Under the assumption that the adversarial loss is minimized for $D_{PR}$ and $\mathcal{L}_{anomaly}$ is minimized for anomaly detection, the resulting embedding $\mathbf{h}_{main}$ is invariant to variations in $\mathbf{h}$ predictive of PageRank:*

$$I(\mathbf{h}_{main}; y^{PR}) \to 0, \tag{18}$$

*where $I(\cdot; \cdot)$ denotes mutual information. Consequently, $\mathbf{h}_{main}$ contains primarily anomaly-related information.*

*Proof.* The training objective for $\mathbf{h}_{\text{main}}$ involves the gradient reversal layer (GRL), which effectively reverses the gradient of $\mathcal{L}_{\text{adv}}$ during backpropagation. Formally, this can be expressed as a min-max optimization:

$$\min_{\mathbf{h}_{\text{main}}} \max_{D_{\text{PR}}} \mathbb{E}\Big[\text{MSE}(D_{\text{PR}}(\mathbf{h}_{\text{pr}}), y^{\text{PR}})\Big]. \tag{19}$$

At the Nash equilibrium of this game: 1. $D_{\text{PR}}$ is optimal in predicting PageRank from $\mathbf{h}_{\text{pr}}$. 2. $\mathbf{h}_{\text{main}}$ receives reversed gradients that penalize any information predictive of $y^{\text{PR}}$.

Since $\mathbf{h}_{\text{main}}$ is updated to minimize the ability of $D_{\text{PR}}$ to predict PageRank, the mutual information $I(\mathbf{h}_{\text{main}}; y^{\text{PR}})$ is minimized. In the ideal case, this mutual information converges to zero, implying that $\mathbf{h}_{\text{main}}$ contains negligible PageRank-related information while preserving anomaly-related signals through $\mathcal{L}_{\text{anomaly}}$.

This proves that the adversarial mechanism achieves feature disentanglement under the stated assumptions. □

# D ALGORITHMIC DETAILS

Algorithm 1 summarizes the training procedure of DIDAN. It combines TIP-based feature extraction, anomaly feature synthesis via a flow-based generator, and adversarial decoupling for robust anomaly detection on dynamic graphs.

# E ADDITIONAL EMBEDDING VISUALIZATIONS

To complement the visualization in Section 4.5, we provide PCA-based 3D projections for the remaining six datasets. For each dataset, the left subfigure shows embeddings without influence disentanglement (w/o ID) and the right subfigure shows embeddings with ID (w/ ID). Purple nodes indicate anomalies, yellow nodes indicate normal nodes.

---

**Algorithm 1** Training Procedure for DIDAN Framework

---

1: **Input:** Dynamic graph sequence $\mathcal{G} = \{G^{(1)}, \ldots, G^{(T)}\}$, node features $\mathbf{X}^{(t)}$, partial anomaly labels, hyperparameters $K, \alpha, \lambda_{\text{adv}}$
2: **Output:** Learned model parameters $\theta$
3: Initialize model parameters $\theta$
4: **for** each time step $t = 1$ to $T$ **do**
5:    **TIP Feature Extraction:** Extract node features $\mathbf{X}^{(t)}$
6:    Compute initial embeddings via MLP: $\mathbf{H}_0^{(t)} \leftarrow \text{MLP}(\mathbf{X}^{(t)})$ Eq. 1
7:    **for** $k = 1$ to $K$ **do**
8:       Propagate node features: $\mathbf{H}_k^{(t)} \leftarrow (1 - \alpha)\hat{\mathbf{A}}\mathbf{H}_{k-1}^{(t)} + \alpha\mathbf{H}_0^{(t)}$ Eq. 2
9:    **end for**
10:    Set node representation: $\mathbf{H}^{(t)} \leftarrow \mathbf{H}_K^{(t)}$
11: **end for**
12: **AFS Anomaly Feature Synthesis:** Collect known anomaly features $\mathcal{D}_a$
13: Train flow-based model $f_\theta$ on $\mathcal{D}_a$ to maximize log-likelihood (see Eq. 5 in the main text)
14: Sample synthetic anomalies: $\mathbf{x}_{\text{gen}} = f_\theta^{-1}(\mathbf{z}), \mathbf{z} \sim \mathcal{N}(\mathbf{0}, \mathbf{I})$ Eq. 3
15: Augment training set with $\mathbf{x}_{\text{gen}}$
16: **for** each training epoch **do**
17:    **for** each batch of nodes **do**
18:       Obtain node representation $\mathbf{h}$ from TIP
19:       Project to influence subspace: $\mathbf{h}_{\text{pr}} = \mathbf{W}_p\mathbf{h}$ Eq. 6
20:       Back-project to remove influence: $\mathbf{h}_{\text{main}} = \mathbf{h} - \mathbf{W}_b\mathbf{h}_{\text{pr}}$ Eq. 6
21:       Compute anomaly predictions from $\mathbf{h}_{\text{main}}$
22:       Compute adversarial influence predictions from $\mathbf{h}_{\text{pr}}$
23:       Compute losses: $\mathcal{L}_{\text{adv}}$ Eq. 7, total loss $\mathcal{L} = \mathcal{L}_{\text{anomaly}} + \lambda_{\text{adv}}\mathcal{L}_{\text{adv}}$ Eq. 8
24:       Update model parameters $\theta$ by backpropagation
25:    **end for**
26: **end for**

---

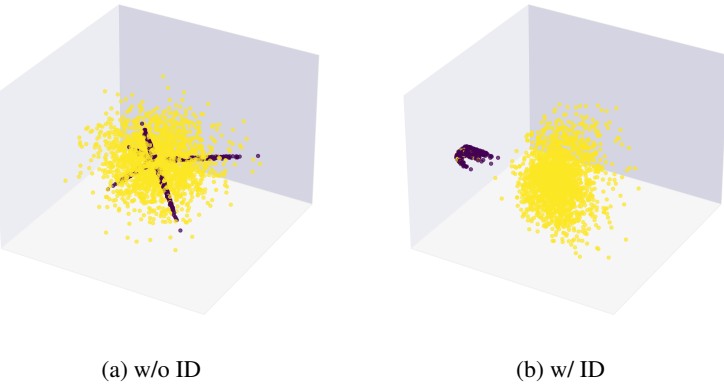

(a) w/o ID                              (b) w/ ID

Figure 5: PCA-based 3D visualization of node embeddings on the EU-Core1 dataset.

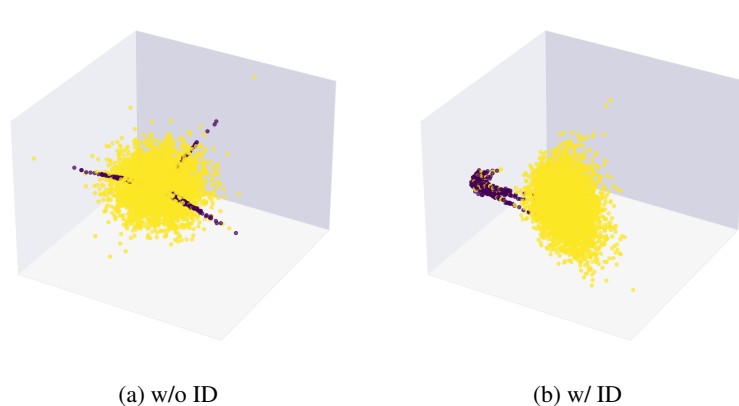

(a) w/o ID                     (b) w/ ID

Figure 6: PCA-based 3D visualization of node embeddings on the ALPHA dataset.

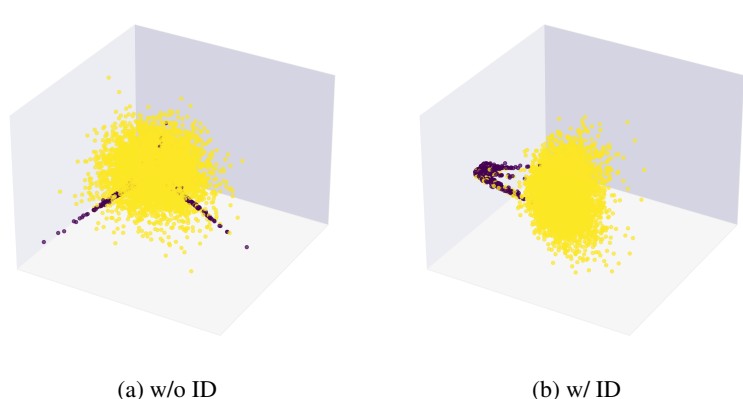

(a) w/o ID                     (b) w/ ID

Figure 7: PCA-based 3D visualization of node embeddings on the OTC dataset.

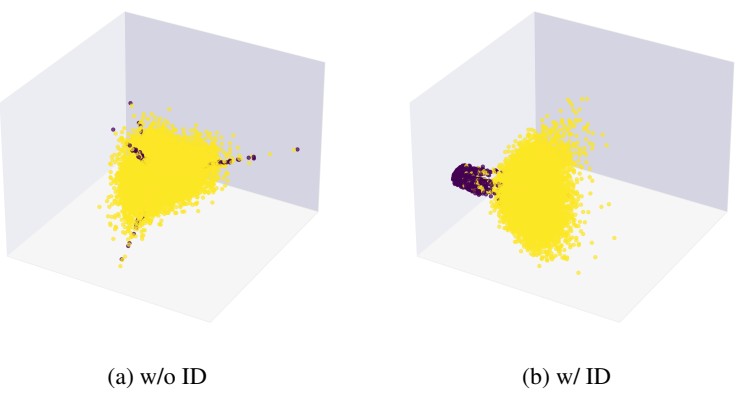

(a) w/o ID                     (b) w/ ID

Figure 8: PCA-based 3D visualization of node embeddings on the Wikipedia dataset.

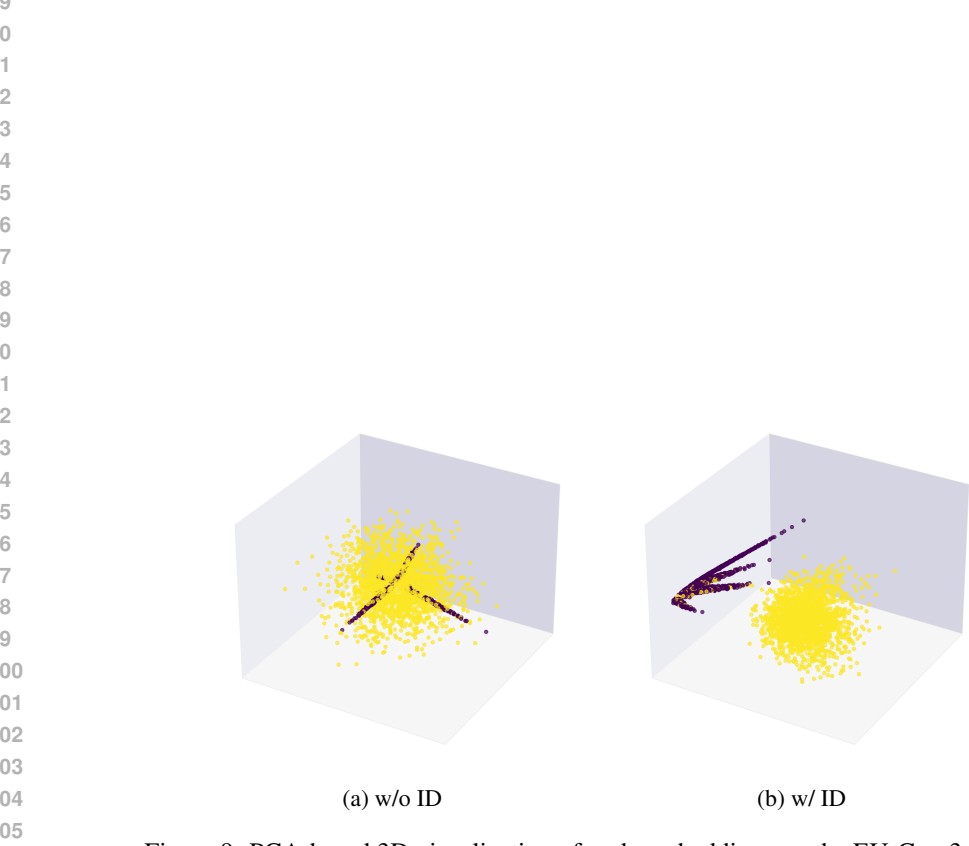

(a) w/o ID                              (b) w/ ID

Figure 9: PCA-based 3D visualization of node embeddings on the EU-Core3 dataset.

