# OpenReview forum: "Beyond Influence: Decoupled Representation Learning for Dynamic Graph Anomaly Detection"
_ICLR.cc/2026/Conference — ICLR 2026 Conference Withdrawn Submission_

### Official Review · Reviewer_Upkr · 2025-10-27

**Soundness:** 2
**Presentation:** 1
**Contribution:** 2
**Rating:** 2
**Confidence:** 4

**Summary:**

This paper proposes DIDAN, a framework for anomaly detection in dynamic graphs. The central motivation is that existing methods often "conflate" benign variations in node influence (e.g., changing PageRank) with true anomalous behavior, leading to high false positives. DIDAN aims to solve this by explicitly decoupling influence-related features from anomaly-related features. The framework consists of three main components: (1) a Temporal Information Propagator (TIP), which uses APPNP to learn stable node representations; (2) an Anomaly Feature Synthesizer (AFS), a flow-based model to generate synthetic anomalies and address class imbalance ; and (3) an Adversarial Influence-Decoupled Detector (AIDD), which uses adversarial training with a gradient reversal layer to remove PageRank-correlated information from the final node representations. Experiments on several real-world datasets show that DIDAN outperforms existing state-of-the-art methods.

**Strengths:**

1. Well-Motivated Problem: The problem of distinguishing benign dynamic changes (like natural fluctuations in node importance) from genuine anomalies is a significant and practical challenge in dynamic graph analysis.


2. Strong Empirical Results: The model achieves state-of-the-art or near-state-of-the-art performance across all reported metrics on multiple benchmark datasets, with particularly large gains on datasets like Reddit and Wikipedia (e.g., +27.73% ROC-AUC on Reddit)

**Weaknesses:**

1. The primary weakness of this work is its lack of methodological novelty. The proposed DIDAN framework appears to be a "stitching" of three independent and well-established techniques, each addressing a separate, common problem in graph anomaly detection.
- TIP: This module is a direct application of APPNP (Predict then Propagate) , a well-known model from 2018. Using APPNP for feature smoothing and propagation is standard practice and not a novel contribution.
- AFS: This module uses a "flow-based generative model"  to alleviate class imbalance. Using generative models (like GANs or flows) for data augmentation in imbalanced settings is a textbook technique. In fact, one of the paper's own baselines, MAMF, also employs "GAN-based anomaly augmentation", which makes this component's contribution feel marginal.
- AIDD: This module uses a projection/back-projection architecture with a gradient reversal layer (GRL) and an adversarial loss. This is the classic mechanism from Domain-Adversarial Neural Networks (DANN) (Ganin et al., 2016) used for feature decoupling. The paper's own related work section (2.2) points to other works using feature decoupling

2. The core premise of the paper—and its main purported contribution—is the decoupling of "influence" from "anomalies." However, the methodology for doing so rests on a critical and likely incorrect assumption.
The paper's design assumes that "temporal variations in a node's PageRank primarily reflect normal influence changes". Based on this, the entire AIDD module is designed to make the final node representation h_main invariant to any information that could predict PageRank.
This assumption is fundamentally flawed. In many real-world scenarios, a sudden and drastic change in PageRank (or other influence-centrality metrics) is precisely the signal of the anomaly itself.

3. No Validation of the Core Claim: The paper claims to solve the problem of benign influence changes being misclassified. To prove this, the experiments would need to show a qualitative analysis or case study of high-influence benign nodes that other methods (like MAMF or SAD) flag as anomalous but DIDAN correctly identifies as normal. The paper provides no such evidence.

4. Given the flaw identified in Weakness 2, a crucial experiment is missing: how does DIDAN perform on a synthetic or real dataset where the primary anomaly signal is a sudden PageRank spike? The current experiments do not test this, and I suspect the model would fail to detect such anomalies by design.

**Questions:**

See Weakness.

---

### Official Review · Reviewer_H4HH · 2025-10-31

**Soundness:** 2
**Presentation:** 2
**Contribution:** 2
**Rating:** 2
**Confidence:** 4

**Summary:**

This paper focuses on solving a meaningful task: anomaly detection in dynamic graphs. This paper proposes DIDAN, which contains three main modules. The authors have claimed that they have achieved remarkable improvement (nearly 100% accuracy on two classic datasets); however, I think the results are unreliable after reading the codes (see weakness for details). Thus, I suggest not accepting this paper. Or maybe I have a wrong understanding about it, where the authors could explain it in the discussion period. I would modify my rating accordingly.

**Strengths:**

- This paper aims to solve a meaningful task.
- The structure of this paper is complete.
- The authors have provided the completed codes, which would promote the development of this field.

**Weaknesses:**

- The authors claim that the DIDAN achieves an F1-score of 0.9873 and 0.9963 on the datasets Wikipedia and Reddit. It means that the detector would hardly make a wrong prediction, which is amazing and unbelievable. Thus, I carefully read the code provided by the authors (it's a good thing that the authors provide the code, which could promote the development of this field). Unfortunately, the improvement/remarkable performance may stem from the incorrect metric calculation (Maybe I have a wrong understanding about it, where the authors could explain it in the discussion period. I would modify my rating accordingly).
  - the authors set drop_last of the test_loader, which would removes some instances in the test set (If the number of instances is not divisible by the batch size).
  - The authors calculate the metrics by averaging the metrics of each batch, where the correct way is to obtain all predictions and labels and calculate the metrics in one time. The two ways could obtain the same value when calculating accuracy, but thet are different when calculating f1-score and roc_auc.
Thus, it leads to that the comparasions between DIDAN and baselines are not fair, where the results are unrealiable.
- The authors do not provide sufficient experimental results to prove the effectiveness of DIDAN and each novel proposed module.
  - In fig.2, replacing APPNP with GCN is meaningless, since APPNP is not the contribution of this paper, and is an existing work. Meanwhile, should the shorten of ID be AIDD?
  - In fig.3, experimenting with different learning rates is meaningless, as the contribution of this paper is unrelated to the learning rate (it does not explore convergence speed, etc.). Exploring the impact of hyperparameters related to DIDAN (such as the trade-off hyperparameter in eq. 8) on performance is more meaningful.
  - In fig.4, after dimensionality reduction, w/ ID does indeed make the blue and orange dots appear separated, but there are still many orange dots within the blue (this may conflict with the nearly 100% performance). A better way is to provide some quantification results of the original (non-dimensionality reduction) representation.
Besides, I think the authors should provide more experiments/results which are highly related to the novelties.
- The authors provide duplicate references (lines 575-582, 598-602)

**Questions:**

pls refer to the weakness above.

---

### Official Review · Reviewer_B8Ro · 2025-10-31

**Soundness:** 2
**Presentation:** 2
**Contribution:** 3
**Rating:** 4
**Confidence:** 3

**Summary:**

The manuscript proposes a framework DIDAN for dynamic graph anomaly detection. The central contribution is the identification and mitigation of entanglement between benign node influence dynamics and genuine anomaly signals. The work posits that fluctuations in metrics like PageRank are often misclassified as anomalies by existing methods. Their proposed solution integrates three components: a temporal propagator for stable representations, a flow-based anomaly synthesizer to address class imbalance, and an adversarial detector to explicitly decouple influence-related features from anomaly-related features.

**Strengths:**

S1. The paper identifies a challenge in dynamic graph anomaly detection: the entanglement of benign node influence fluctuations (like PageRank changes) with genuine anomalous signals.

S2. The authors have made their code publicly available, that allows for verification and builds in the results.

**Weaknesses:**

W1. The paper's entire premise rests on the claim that existing methods "conflate... benign variations with anomalous behaviors." However, the experiments do not provide direct evidence for this claim or that the proposed method specifically solves it.

W2. In Section 3.4, the authors "assume that temporal variations in a node's PageRank primarily reflect normal influence changes." Does this assumption have flawed? An anomalous event (like, a coordinated bot attack, a "pump and dump" scheme in a transaction graph) also could cause a massive and sudden spike in a node's PageRank. It is recommended authors defend this assumption.

W3. The framework appears to be a "patchwork" of existing, well-known components: TIP: direct application of APPNP (Gasteiger et al., 2018); AFS: standard application of a flow-based generative model; AIDD, like DANN, Ganin et al., 2016.
The novelty is not in the components, but in their combination to solve the problem of influence-entanglement. This is a valid engineering contribution, but for ICLR, the novelty is not enough.

W4. It is recommended authors discuss the method’s limitations and proposing specific directions for future research in conclusion section.

**Questions:**

Q1. The AIDD module uses linear projection W_p and back-projection W_b to isolate and remove the influence-correlated components. Is a linear projection sufficient to capture what might be a complex, non-linear relationship between a node's high-dimensional embedding h and its influence score?

Q2. The paper assumes that temporal variations in PageRank primarily reflect normal influence changes. However, couldn't a class of anomalies be specifically designed to manipulate this metric? By adversarially training the detector to ignore PageRank-correlated features, is there a risk of creating a blind spot?

Q3. What was the empirical or theoretical rationale for selecting PageRank over other dynamic influence or centrality measures, such as betweenness centrality or simple degree?

---

### Official Review · Reviewer_1qhj · 2025-11-02

**Soundness:** 3
**Presentation:** 3
**Contribution:** 3
**Rating:** 4
**Confidence:** 4

**Summary:**

This paper addresses the overlooked issue of influence–anomaly entanglement in dynamic graph anomaly detection (DGAD). Existing DGAD methods often mistake natural variations in node influence (e.g., PageRank changes) for anomalies, resulting in false positives. The proposed framework, DIDAN (Dynamic Influence-Decoupled Anomaly Network), explicitly disentangles node influence dynamics from true anomaly signals. It consists of three modules: 1. Temporal Information Propagator (TIP); 2. Anomaly Feature Synthesizer (AFS); 3. Adversarial Influence-Decoupled Detector (AIDD). Experiments on seven dynamic graph benchmarks show large performance gains.

**Strengths:**

Three strong points are listed as follows:

1. The paper identifies and formalizes a key limitation in DGAD—confounding between influence dynamics and anomaly signals—which prior works largely ignored. The explicit influence–anomaly decoupling perspective is original and practically relevant for evolving network systems.

2. DIDAN’s modular architecture (TIP, AFS, AIDD) is conceptually coherent: TIP ensures representation stability, AFS balances classes via generative modeling, and AIDD enforces disentanglement. The theoretical analysis (convergence proofs and bijectivity guarantees) strengthens credibility.

3. The experiments cover diverse datasets with meaningful ablation studies, parameter sensitivity tests, and embedding visualizations that clearly demonstrate the impact of decoupling. DIDAN consistently outperforms all baselines across metrics, showing robustness and generalizability.

**Weaknesses:**

Some weak points are listed as follows:

1. While the anomaly decoupling idea is new, the building blocks (APPNP propagation, flow-based generative model, GRL-based adversarial training) are adaptations of established methods. The framework’s originality lies in integration, not algorithmic invention. Please elaborate on this.

2. The paper claims “efficiency for sparse graphs” (theoretical complexity of $O(T (Nd^2 + KEd) + BMd^2 + SLd^2)$), but empirical validation of runtime or memory efficiency on large-scale graphs is missing.

**Questions:**

Please see the weak points.

---

### Comment · Area_Chair_7YJH · 2025-11-28

Dear Reviewers,

Thank you for your valuable time and expertise in reviewing this paper.

The authors have now submitted their rebuttal. We would appreciate it if you could review their responses and assess whether your concerns have been addressed, if you haven't done this.

Best regards,

AC

---

### Note · Authors · 2025-12-05

I have read and agree with the venue's withdrawal policy on behalf of myself and my co-authors.